# Participant Experience of a Modified Sports Program— A Curriculum Investigation in Gaelic Games

Kevin Gavin [1,2,*], Jamie Taylor [1,2,3], Stephen Behan [1,2], Peter Horgan [4] and Áine MacNamara [1]

1   School of Health and Human Performance, Faculty of Science and Health, Dublin City University, D09 W6Y4 Dublin, Ireland; jamie.taylor@dcu.ie (J.T.); stephen.behan@dcu.ie (S.B.); aine.macnamara@dcu.ie (Á.M.)
2   Insight SFI Research Centre for Data Analytics, Dublin City University, D09 W6Y4 Dublin, Ireland
3   Grey Matters Performance Ltd., Stratford upon Avon CV37 9TQ, UK
4   Gaelic Athletic Association, Croke Park, D03 P6K7 Dublin, Ireland; peter.horgan@gaa.ie
*   Correspondence: kevin.gavin@dcu.ie

**Abstract:** Modified sports programs aim to encourage children's participation in sport and develop the skills required for future participation, with existing research supporting their positive influence on participants' enjoyment, skill performance, and learning. However, limited research in this area and potential difficulties in contextual application underscore the need to understand stakeholders' perceptions and the dilemmas of practice. Therefore, this study aimed to explore stakeholders' perceptions of the Gaelic games modified sports program, Go Games, utilising the intended–enacted– experienced curriculum model as a framework. Short semi-structured interviews were conducted with 180 participants, including players ($n = 92$), parents ($n = 62$), and coaches ($n = 26$). Data were analysed using qualitative content analysis. The findings indicate a strong coherence between the experiences and perceptions of coaches and parents with the intended curriculum, but a disparity in understanding the purpose and objectives of modifications amongst parents. This study underscores the role of enjoyment for participants, but also highlights the high variability in the sources of this enjoyment. Prominently, coaches faced a range of dilemmas of practice based on the need to juggle often competing sources of motivation and enjoyment. This study suggests the need for greater parental understanding and significant support for coaches to manage these dilemmas of practice.

**Keywords:** modified sports program; Gaelic games; youth sport; coaching; curriculum





## 1. Introduction

Sport policies allow for national governing bodies (NGBs) to influence practices and provide direction for local community sports clubs [1]. While sport policies have historically prioritised competitive sport and elite performance, there has been a growing interest in lifelong participation in sport and physical activity (e.g., [2,3]). Reflecting this, Collins et al. [4] suggested a consideration of different motivations for participation and achievement across a continuum of elite-referenced excellence, participation for personal excellence, and participation for personal well-being. The potential for lifelong participation alongside high performance objectives suggested by the Three Worlds model has been reflected in a variety of national sport organisation policies [1,5]. Thus, there has been an increasing drive for NGBs to implement policies aimed at meeting the needs of participants across the motivational spectrum, enhancing inclusion, retention, and quality of experience, most often aimed at younger participants [1].

While sport participation is the most popular form of leisure-time physical activity amongst children and adolescents worldwide [6], extensive research indicates that participation peaks at 10-14-years-old and declines throughout adolescence [7,8]. Among a range of intrapersonal, interpersonal, organisational, environmental, and policy factors that mediate sport participation, a lack of physical competence, and enjoyment have

emerged as key determinants of dropout [9]. Consequently, there is growing discourse about how to shape early experiences to increase the longer-term participation of children in sport [10]. One such concern is that the youth sport milieu has tended toward emphasising winning and competition at the expense of facilitating the inclusion and retention of participants [11–13]. Furthermore, a point of contention exists within the literature regarding how the coach shapes their practice to provide participants with the foundation for participation in lifelong physical activity [14]. One perspective argues that meeting the needs of children involves maximising fun, engagement, and participation as the primary drivers of youth sport provision [15,16]. Another perspective emphasises the importance of systematically developing perceived and actual competence facilitated through appropriate feedback, instruction, and organisation provided by coaches [17]. Of course, it is unlikely that the most effective early experience for young people in sport is going to be an 'either-or' decision, but instead it is important to consider the balance of experience that most effectively supports long-term participation [4].

### 1.1. Modification and Scaling in Youth Sport

In response to this challenge, a number of NGBs have employed modified sport programs to encourage more children to participate in organised sport and to support the development of the necessary fundamental motor- and sport-specific skills [18,19]. Examples of these policy level changes include the 'AusKick' program in Australian rules football [20], the 'Tennis10s' program in tennis [21], and the 'NetSetGo' program in netball [22]. In each of these instances, the target sport is modified through changes to game rules, equipment, and/or playing area based on the developmental capacities of participants with a focus on the development of actual and perceived competence, motor skill acquisition, inclusion, and maximising participation [19,23]. The evidenced benefits of scaling rules and equipment include greater engagement, enjoyment, and improved skill performance and learning [23]. In addition, modifications have been linked to increased sport participation in 5–10-year-old children [19,24]. Consequently, it is likely that the scaling of rules and equipment may support the development of the actual and perceived competence shown to be necessary elements for long term participation [14].

### 1.2. Competition in Youth Sport—Moving beyond Equipment and Dimensions

One modification that has received considerable attention is the role of competition in youth sport [25]. Much of the recent literature has suggested the need for sporting organisations to de-emphasise both competition and winning to promote developmental goals (e.g., [26,27]). Instead of emphasising outcome (e.g., winning and losing), removing league or competition structures and including regulations ensuring that all players, irrespective of ability, have equal playing time is hypothesised to increase involvement [28]. Furthermore, Burton et al. [29] suggest that modifying the competition structures (e.g., leagues) creates competitive experiences that better align to the wants and needs of young players. In response, some sport organisations have made significant policy changes in an attempt to de-emphasise competition, such as discouraging the formation of competitive teams, the keeping of scores or league standings, and ensuring equal playing time at earlier ages [10,12]. In contrast, Torres and Hager [13] argue that the prevailing trend among youth organisations to excessively de-emphasise or eliminate competition is "unwarranted and misleading for children" (p. 194) and argue that rather than eliminate the "central purpose" of competitive sport, youth sport should aspire to teach children to "compete in a good and decent manner" (p. 207). While Torres and Hager's [13] endorsement of fostering positive competition is commendable, their promotion of competition as the central purpose of youth sport does not adequately address the diverse range of interests and motivations amongst youth players, such as enjoyment, skill development, and social interaction [30]. Furthermore, it is important to acknowledge the considerable variation in cognitive, social, and physical maturity among children and adolescents, which inherently influences their participation motivation [31,32]. Nevertheless, this raises an important ques-

tion regarding the extent to which youth sport differentiates and accommodates varying participant motivations.

Competition has been defined as "a social process that occurs when rewards are given to people on the basis of how their performances compare with the performances of others doing the same task or participating in the same event" [33]. Competitiveness, on the other hand, has been defined as "a disposition to strive for satisfaction when making comparisons with some standard of excellence in the presence of evaluative others in sport" [34]. In this sense, it appears that there is a distinction to be drawn between coaches and participants focusing on winning as the sole outcome, and the potential benefits that may accrue for young athletes competing against their peers. Simply, it is possible that both winners and losers can derive value from competing [13]. This may in turn present a complex problem for the coach, one that requires the ability to consider stakeholder perspectives about the design and structure of competition in youth sport and move beyond the dichotomy of competitive and non-competitive youth sport.

### 1.3. Curriculum

This focus on enhancing the experience of youth in sport can be seen through a curriculum lens, whereby the curriculum represents all elements of the participant's experience [33]. To better understand the current state of a modified sports program, we can adopt and employ the intended–enacted–experienced curriculum model from the educational domain which separates the curriculum into 'lenses' (e.g., [34]). The intended curriculum "establishes the curricular goals, learning outcomes, or national standards explaining what students should be able to know and do after completing the curriculum" (p. 2) [35]. The enacted curriculum "is how instructors translate the intended curriculum into teaching and assessment in actual courses" (p. 2) [35]. The experienced curriculum "is the curriculum perceived by students in response to instruction" (p. 2) [35]. In the context of participation in youth sport, the intended curriculum involves multiple levels ranging from the national policies that shape practice, to the planning of coaches at the local level which establishes the goals and learning outcomes of participants. The enacted curriculum is what coaches do when 'delivering' the intended curriculum. Finally, the experienced curriculum is what sporting participants actually experience when engaging with these intentions and actions. This model allows for a comprehensive investigation of a modified sports program, considering not just the intended goals, but also the coherence of these objectives compared to coaching practice and participant experience.

### 1.4. Go Games: An Example of a Modified Sports Program

Gaelic games, inclusive of Gaelic football, hurling and camogie, are field-based invasive team sports indigenous to Ireland [36]. Gaelic games are governed by the Gaelic Athletic Association (GAA), Ladies Gaelic Football Association (LGFA), and Camogie Association. Despite some differences between the playing rules, the primary distinction between the codes lies in the equipment and specific skills employed [37]. In Gaelic football, played by both males and females, the ball is comparable to that used in soccer, and requires players to perform several skills, such as hand passing, kick passing, catching, blocking, and tackling [38,39]. In comparison, both the male game of hurling and the female game of camogie is played with a stick called a "hurley" and a small leather ball called a "slíotar", with players performing several skills, including catching, blocking, and striking the slíotar with the hurley, either from the hand or on the ground [40].

As Ireland's largest community and sporting organisation, Gaelic games play a pivotal role in Irish society, with almost 3000 affiliated clubs, approximately one million members, 500,000 players, and 100,000 coaches [41]. A significant example of a policy level change to the intended curriculum was the introduction of a modified sports program for children aged 7 to 11 years, entitled "Go Games", with the aim to "provide children with an appropriate introduction to competition on a phased basis, while also providing a sufficient skill development challenge" [42]. Specifically, the Go Games are underpinned

by several principles, including ensuring "everyone has a go" (i.e., equal playing time for all participants), emphasising skill development, promoting enjoyment, and refraining from keeping or publishing scores of the games or announcing winners [42]. The Go Games format incorporates modifications of the rules, equipment, playing area, and player numbers of Gaelic games, with slight variations across the different age grades to meet the developmental needs of participants. For example, at the under-10 age grade, games are played on a 100 × 45 m pitch with a smaller sized ball, featuring a maximum of nine players per team. Additionally, players are only permitted one touch of the ball before passing or shooting [42].

In the context of Gaelic games, the intended–enacted–experienced curriculum model serves as a multi-level framework to consider the impact of the Go Games modified sports program and coherence of policy, practice, and experience [35]. Of course, manipulating and scaling the rules, experience, and dimensions of youth sport has considerable face value and few people would argue against age- and stage-appropriate versions of sport. Although, as described by Buszard and colleagues [43], modified sports programs are often seen as an entry level strategy to attract children into a sport and maintain participation. However, less is understood about key stakeholders' perceptions of modified and scaled versions of competition structures in youth sport. Therefore, this study sought to explore players', parents' and coaches' perceptions of the Go Games modified sports program, utilising the intended–enacted–experienced curriculum model as a framework. By investigating a multitude of experiences, this study aimed to gain a deeper understanding of the lived curriculum experience and the challenges associated with the modified sports program.

## 2. Materials and Methods

### 2.1. Design and Methodology

This study was guided by a pragmatic research philosophy, driven by a commitment to generating practical and impactful knowledge that can contribute to real-world outcomes and positively influence people and practices [44]. Rather than being guided by a distinct epistemological approach, a pragmatic research philosophy focuses on application and usefulness to practitioners, offering feasible and actionable measures in real-world settings [45]. Pragmatic approaches also emphasise the prioritisation of questions and methods that are practically meaningful rather than seeking generalisable truths or subjective constructions [46]. To reflect this pragmatic approach and our research objectives, a qualitative methodological approach, involving short, one-to-one semi-structured interviews, was employed. This qualitative approach facilitated a comprehensive exploration of the perceptions and experiences held by players, parents, and coaches, offering an understanding of their perspectives on the modified Gaelic games programme.

### 2.2. Participants

A total of 180 participants, representing players, parents, and coaches, were purposefully sampled from eight Gaelic games clubs across four different counties to participate in this study. All participants were recruited from an evenly distributed geographical sample to provide a nationally representative spread of participants across the country. Specifically, we sought to invite participants from both the male codes of Gaelic football and hurling, and the female codes of ladies Gaelic football and camogie. Moreover, we aimed to recruit participants from across the under 10, under 11, and under 12 age grades. These age grades were selected due to the slight variations in their organisational structure, as both the under 10 and under 11 age grades are organised on a blitz basis, which, in the context of Go Games, entails the participation of a minimum of three clubs or multiple teams from two clubs. In contrast, the under 12 age grade is organised on both a league and a blitz basis.

### 2.2.1. Players

A total of 92 players volunteered to participate; 48 players from the under 10 age grade (male: $n = 16$; female: $n = 32$), 14 players from the under 11 age grade (male: $n = 8$; female: $n = 6$), and 30 players from the under 12 age grade (male: $n = 16$; female: $n = 14$).

### 2.2.2. Parents

A total of 62 parents of the player participants also volunteered to participate; 26 parents of children at the under 10 age grade (male: $n = 10$; female: $n = 16$), 12 parents of children at the under 11 age grade (male: $n = 7$; female: $n = 5$), and 24 parents of children at the under 12 age grade (male: $n = 13$; female: $n = 11$).

### 2.2.3. Coaches

A total of 26 coaches, who were part of the coaching teams for the player participants, volunteered to participate; 14 coaches from the under 10 age grade (male: $n = 7$; female: $n = 7$), 5 coaches from the under 11 age grade (male: $n = 3$; female: $n = 2$), and 7 coaches from the under 12 age grade (male: $n = 2$; female: $n = 5$).

### 2.3. Data Collection

Prior to data collection, ethical approval was granted from the authors' Institutional Ethics Committee (#2023/135), and the first author visited each 'Go Games' team to meet with coaches and share study information with the parents. Informed consent was ascertained from interested coaches and parents, while informed assent was also obtained from parents on behalf of their children. Guided by the exploratory nature of this study and based on the pragmatic approach [44], different semi-structured interview guides were developed for players, parents, and coaches. All consisted of open-ended questions that revolved around various perceptions of the Go Games modified sports programme (i.e., Do you know the score of the games today? Do you know who won the games today? How competitive are Go Games? Do you/your child/your players enjoy playing in Go Games? What are the best aspects of Go Games? Is there anything you would change to improve Go Games?). In addition, follow-up probes and prompts were developed to allow elaboration on key points and promote consistency across participants (e.g., Does winning matter to you/your child/your players? What made it enjoyable/not enjoyable? What made it competitive/not competitive?) [47].

All the player and parent participants completed the short semi-structured interviews at the Gaelic games club where the Go Games were taking place. Players completed their interviews immediately after the conclusion of their game, with their parents present, while parents were interviewed during their child's game. This timing was deliberately selected to maximise the ecological validity of this study, enabling participants to share their immediate reactions to the Go Games program. To encourage comfort and openness, interviews took place away from other individuals at the side of the Gaelic games field. All coach participants completed the short semi-structured interviews remotely at a time that was convenient to them. All the interviews were recorded using a digital voice recorder (Philips VoiceTracer; Philips Electronics UK Ltd., Guildford, England), with an average interview duration of twelve minutes for coaches, five minutes for parents, and four minutes for players.

### 2.4. Data Analysis

Following data collection, all interviews were transcribed verbatim, then checked against recordings for accuracy. The first author then grouped all transcripts by participant group (player, parent, coaches) and uploaded them into NVivo 12 software (QSR International Pty Ltd., Doncaster, VIC, Australia) which facilitated the organisation and coding of the dataset. Qualitative content analysis was chosen as the specific analytic strategy to identify patterns of meaning (i.e., themes) in the data and was conducted in two distinct phases: deductive and inductive. One of the major benefits of content analysis

is its flexibility in terms of research design and that the use of deductive and/or inductive methods should be determined by the purpose of the research [48].

### 2.4.1. The Deductive Phase

Deductive content analysis was then performed according to Elo and Kyngäs' [48] description. In the deductive phase, a structured categorisation matrix based on the intended–enacted–experienced curriculum model was constructed. The categorisation matrix was used as a lens during the analysis of the text and to form domains under which the data were sorted. Specifically, this model allowed us to examine how the intended Go Games principles, recommendations, and guidelines (intended curriculum) translated into actual practice (enacted curriculum), and how these experiences were perceived and interpreted by players, parents, and coaches (experienced curriculum). The analysis began with multiple readings of the transcripts to become familiar with the data and acquire an overview of the texts. Next, the transcripts were carefully reviewed for content, and text corresponding to the categorisation matrix was highlighted, coded, and transferred into the relevant description categories in the matrix (see Table 1). The first author took the lead in the analysis, while the other authors appraised and evaluated the positions of the text transferred to the matrix.

**Table 1.** Example of analysis process in deductive phase using the categorisation matrix.

| Intended Curriculum | Enacted Curriculum | Experienced Curriculum |
| --- | --- | --- |
| "I think most people would be of the opinion that it's not just about the best players being on the pitch, it's about everybody being on". (Coach 21) | "The reality is you might have three or four very strong players and they can dominate very easily. So that's a bit of a challenge in that to get the other kids involved". (Coach 14) | "He sees that he's improving, that he's getting better and so he likes it, and he always wants to come back". (Parent 31) |

### 2.4.2. The Inductive Phase

Inductive analysis was performed on the data, guided by the four staged framework described by Bengtsson [49]: (a) decontextualization, (b) recontextualization, (c) categorisation, and (d) compilation. The first stage began with multiple re-readings of the transcripts to facilitate data familiarisation beyond the earlier categorisation of the text in the deductive analysis. Then, each transcript was carefully read, highlighting meaning units by denoting the constellation of sentences or paragraphs that are central and relevant to the purpose of this study. These meaning units were then assigned codes. After coding, the second stage involved re-reading of the meaning units, and emergent codes to confirm or re-code the meaning units. Similar codes were coalesced, and different codes remained distinct. The third stage involved the sorting of codes into sub-themes and themes, which were then checked for internal homogeneity and external heterogeneity (see Table 2). The last stage involved the completion of the analysis and writing up process. The analysis was led by the first author, while the other authors were the co-analysts. Although coding was undertaken by the first author, investigators' triangulation was conducted to validate the process. This was performed by having the second and third authors code a number ($n = 6$) of randomly chosen transcriptions independently and deriving their own sub-themes. The authors had to compare the coding, discussing reasons for developed themes. Differences were solved by revisiting textual data, derived meaning units, and condensed meaning units, and mutually re-coding the data, and agreeing on appropriate sub-themes and themes.

**Table 2.** Example of analysis process in inductive phase.

| Meaning Unit | Code | Sub-Theme | Theme |
|---|---|---|---|
| "I'm in favour of the Go Games, I think, it's a good model. I think the best part of it is they aren't tracking the scores". (Coach 2) | Keeping Score | De-emphasising Scores and Winning | Buy-in to the Go Games Curriculum |
| "I think the streaming worked and the reason why it worked is it gives every player, regardless of skillset and opportunity to play and be involved in the game". (Parent 48) | Streaming | Balancing Equal Participation and Competitiveness | Coaches' Experience Enacting the Go Games Curriculum |
| "Meeting new people and I get to play with my friends". (Player 82) | Enjoyment of Social Interaction | Players Enjoy the Go Games | Player and Parent Experience of the Go Games Curriculum |

## 3. Results

The purpose of this study was to explore players', parents', and coaches' experiences and perceptions of the Go Games modified sports program. Accordingly, the findings are organised into three distinct themes: buy-in to the Go Games curriculum, coaches' experience with enacting the Go Games curriculum, and players' and parents' experience of the Go Games curriculum. Table 3 presents all themes and sub-themes with exemplar quotations to allow the reader to engage with the participants' perceptions and experiences and to illustrate the analysis.

**Table 3.** Themes and sub-themes with exemplar quotations of participants' experiences and perceptions of Go Games.

| Theme | Sub-Theme | Raw Data Exemplars |
|---|---|---|
| Buy-in to the Go Games Curriculum | Equal Participation | "Everybody playing is the most important thing for us, getting them game time. So, adjusting the size of the teams and the pitches and accommodating, making sure we get them game time is important". (Coach 8)<br>"The inclusion of everyone is a major thing. They all get game time and get out in the pitch. It's important to them". (Parent 23) |
| | Emphasis on Enjoyment | "All you want them to do is all play and enjoy it". (Coach 12)<br>"The primary reason to play it should be about enjoyment and walking away with positive attitude". (Parent 5) |
| | Skill Development | "It's more focused on the skills. We're asking someone to do a jab lift or asking them to kick the ball. It's all based on skill development". (Coach 19)<br>"They do the training, they practice with the friends. They're trying to get better, and then when they go out on the field and they can do the things well that they've practiced, and then they just feel like they've mastered something". (Parent 56) |
| | De-emphasising Scores and Winning | "I feel that the Go Games is worth defending definitely. And the philosophy around not keeping score is really important". (Coach 14)<br>"I just think the fact that there's no one really keeping a score or such that they can relax into it and just enjoy it and they're not under pressure to win all the time". (Parent 9) |
| Coaches' Experience Enacting the Go Games Curriculum | Ensuring Enjoyment Among Players | "They enjoy playing for their club, they enjoy playing with their friends, they enjoy competing with other players, even if there's no scoreboard as such. They really get a lot out of putting a jersey on and going off to play another club". (Coach 26) |
| | Balancing Equal Participation and Competitiveness | "We've been involved in very one-sided matches sometimes when we've been on top, but we've made sure we're not going to totally maul the opposition, because we'd be conscious that has happened to us and it's not nice". (Coach 17) |

**Table 3.** *Cont.*

| Theme | Sub-Theme | Raw Data Exemplars |
|---|---|---|
| | Navigating Scores and Winning | "So I focus less on winning and the competition level. The focus isn't on that, which is good, even though the girls probably know whether they have won or lost". (Coach 6) |
| | Procedural Challenges and Consistency | "So at the younger age you might have parents being a bit more vocal towards referee for example". (Coach 22) |
| | Players Enjoy the Go Games | "He does like the competitive side of it, but he does like the social end of it as well because he's hanging around with boys from school and he's got to make new friends". (Parent 34)<br>"I scored. I kind of like it when I score and I did like that we did get a few scores. At least we scored two or three points". (Player 28) |
| Player and Parent Experience of the Go Games Curriculum | Competition and Competitiveness | "The kids keep score. And they have done since they were, I'd say six or seven". (Parent 44)<br>"You can see different standards and different teams, but I suppose there's games where they're probably more evenly matched and you can see that there is a proper element of competition in it". (Parent 16)<br>"We weren't told the score, but I know the winning score. The score was five points to two". (Player 16)<br>"Every time one person went in for the ball, they just went back to the team, was like keep going back and forward. And then goal after goal after goal." (Player 87) |

### 3.1. Buy-In to the Go Games Curriculum

This theme explores the coherence of the experiences and perspectives of coaches and parents with the intended curriculum of Go Games, specifically the core principles and recommendations emphasising equal participation, enjoyment, skill development, and minimal focus on scores or trophies.

### 3.1.1. Equal Participation

A fundamental principle of Go Games is to ensure equal participation amongst players, exemplified by the phrase 'Everyone Has a Go'. Coaches consistently acknowledged the importance of equal participation within Go Games: "I think most people would be of the opinion that it's not just about the best players being on the pitch, it's about everybody being on". Parents echoed these sentiments, regularly emphasising the inclusive nature of Go Games: "I love the Go Games, the format that everyone gets a game, and everyone gets playing time".

### 3.1.2. Emphasis on Enjoyment

The Go Games guidelines place a strong emphasis on creating an environment that optimises enjoyment. The consensus across coaches was that a primary objective of Go Games should be to ensure that children derive enjoyment from their participation: "we have the objective of helping them enjoy Gaelic games". Parents echoed this sentiment, emphasising that the primary motivation behind participation in the Go Games should be enjoyment: "most importantly, it's got to be about them enjoying it". There was also strong agreement among coaches and parents that fostering enjoyment in Go Games is not only about the immediate experience but also about establishing a foundation for continued participation in Gaelic games. Coaches strongly perceived their role as encouraging longer term participation, rather than shorter term aims: "it's about the bigger picture and what we're trying to achieve here with the kids. We want the girls to enjoy their sport, and to hopefully play Gaelic games when they're 17, 18, 19, 20 and beyond". This perspective was echoed by parents, who viewed Go Games as an opportunity to promote continued participation in the sports: "I think with enjoying it, that's where the kind of interest for the sports comes and then in later years, she'll stay with them".

### 3.1.3. Skill Development

The Go Games guidelines place significant emphasis on the development of players' skills. Overall, coaches frequently acknowledged the importance of Go Games in promoting skill development amongst their players: "It's more important about development and they're building all their skills". In particular, coaches were supportive of the Go Games format, highlighting that the small-sided games and modified rules provided players with an opportunity to develop and perform skills: "I think it's extremely important that they are small-sided. The structure allows the players to develop their skills". Parents also recognised the role that Go Games plays in providing players with an opportunity to develop and utilise their skills: "they're learning the skills of the sport and they're getting to use them in the games".

### 3.1.4. De-Emphasising Scores and Winning

A core principle of the Go Games intended curriculum is to ensure that there is no recording or publishing of scores, declaring of winners, or presentation of trophies, cups, or awards. Coaches consistently affirmed their commitment to this principle, emphasising their minimal focus on scorekeeping or winning: "as the coaching group we put zero emphasis on the score". Similarly, parents echoed the sentiment of reduced emphasis on recording scores and winning: "I'm in favour of the Go Games, I think, it's a good model. I think the best part of it is they aren't tracking the scores".

### *3.2. Coaches' Experience Enacting the Go Games Curriculum*

This theme explores the experiences and viewpoints of coaches as they enact the Go Games curriculum, balancing the intended curriculum with the practicalities and challenges they encounter.

### 3.2.1. Ensuring Enjoyment among Players

Coaches consistently expressed their belief that players enjoyed participating in the Go Games: "they definitely enjoy it. Either through feedback from the parents or seeing it from the kids themselves, they enjoy the Go Games". However, coaches also acknowledged that enjoyment manifests differently among individual players, with social interaction and camaraderie, skill development and competence, and competitiveness emerging as prominent sources of enjoyment for players. As one coach stated: "It's going to be different, people will get different enjoyment from it. For some of them they're just happy to be out with their friends. For some of them it's a chance to improve. And some of them love playing against other teams".

Additionally, coaches acknowledged the complexity of catering to this diverse spectrum of enjoyment among the players. For example, one coach outlined how they actively promoted fostering a sense of camaraderie among players: "It's very important for them that they know the others on their team, that there's a friend there with them so we spend some time on that". Similarly, another coach articulated how they promoted competence by frequently encouraging players to attempt specific skills during Go Games: "We ask them to do two, three things, like marking up, heads up, taking shots. That's what we would concentrate on".

### 3.2.2. Balancing Equal Participation and Competitiveness

Coaches demonstrated a commitment to ensuring that all players receive equal playing time, and they often discussed the flexibility within the Go Games guidelines to adapt to create opportunities for every player to participate fully: "everybody playing is the most important thing for us. So, adjusting the size of the teams and the pitches and accommodating, making sure we get them game time is important". Nevertheless, coaches reflected that this commitment to equal participation often resulted in unevenly matched teams, which negatively impacted the competitiveness of the games: "the reality is you

might have three or four very strong players and they can dominate very easily. So that's a bit of a challenge in that to get the other kids involved".

In response to this, coaches perceived that maintaining competitiveness was critical to enhancing player experience, something often achieved by streaming teams based on individual ability: "as coaches, we've got a responsibility to make sure the games are competitive. We want to keep the games even". The practice of streaming was strongly supported amongst coaches although not explicitly intended by the NGB: "I think the streaming worked and the reason why it worked is it gives every player, regardless of skillset and opportunity to play and be involved in the game". Streaming was viewed as particularly important for players who coaches considered to have lower skill levels or ability: "I think we've seen a very positive result from it this year. I suppose it helps weaker or less confident players to come on if they're playing with people of similar ability".

### 3.2.3. Navigating Scores and Winning

While the Go Games guidelines strongly discourage the recording and publication of scores or declaring of winning teams, coaches acknowledged the reality that players are often aware of game outcomes: "Yeah, they know if they have won themselves. Regardless of whether we tell them or not. They'll figure it out". Coaches noted heightened interest in the outcomes of the Go Games amongst the players in the older age groups in particular: "when they're getting to the age of 12, they know themselves whether they've won or lost". Despite this, coaches underscored their commitment to de-emphasising scores and winning amongst their players: "we've made a conscious effort not to put an emphasis on keeping score or winning, we more so put an emphasis on developmental skills and enjoying the games".

Overall, while most coaches stated that they do not track the scores of the games themselves, they would typically be aware of whether their team won or lost: "I genuinely don't count the scores, but I would know if they won or lost. You'd know just by watching it". Nevertheless, a minority of coaches did report recording scores with the intention of monitoring their team's progress to ensure games remained evenly matched: "we would monitor scores. We'd be more inclined to look at the progress that's happening on the pitch and maybe try and shape the teams if it's one-sided". Additionally, some coaches reported tracking scores to inform decisions on how to stream their teams based on player abilities: "We take down the scores for streaming purposes".

### 3.2.4. Procedural Challenges and Consistency

While coaches were largely in favour of the Go Games guidelines and principles, they did encounter procedural challenges in their efforts to implement the intended curriculum. Specifically, despite their support of the flexibility to modify the structure of the Go Games to ensure equal player participation, they did report to encounter challenges when other coaches or clubs did not adhere to this principle: "generally speaking the Go Games are adhered to by the clubs. However, we have occasionally met situations where maybe the other team has two players more than us and they refuse to give a player over. So, we're playing two players down". Furthermore, coaches encountered challenges from opposing coaches in their efforts to uphold the guidelines for the appropriate playing area, player numbers, and the non-recording of scores. "We've come up against teams who want to play a full pitch, want to play 15 players aside, want to keep score". Additionally, coaches outlined that parents of the children are often unaware of the Go Games principles and guidelines, which sometimes resulted in parents shouting from the side lines: "parents have mentioned to me at other matches how aggressive they're finding or parents from the other team".

To address these procedural challenges and promote consistency, coaches expressed a desire for more comprehensive coach education: "I think there's room for delivery of an educational programme. I think if you get coaches to buy into the Go Games and they understand what the ideas behind it are, that's probably how you maintain it and

strengthen it". Moreover, coaches reported a need for, and importance of, communication of the Go Games principles to parents: "I suppose one thing is just that the ethos of the Go Games is known by all and getting that message out to parents. So, I suppose just maybe a greater awareness maybe amongst people of what you're trying to achieve".

*3.3. Players' and Parents' Experience of the Go Games Curriculum*

This theme explores the players' and parents' experiences of the enacted Go Games curriculum.

### 3.3.1. Players Enjoy the Go Games

The intended principle of promoting enjoyment was not only prominently featured in the Go Games guidelines but was also strongly reflected in the experiences of the participants. For example, players were asked to rate their enjoyment on a scale of one (low) to five (high), with almost all players rating it a five. Parents echoed these sentiments, emphasising that their children enjoy participating in the Go Games. As one parent stated: "Yeah, absolutely. She loves the Go Games". A significant number of players emphasised social interaction and camaraderie as key factors contributing to their enjoyment of Go Games. Players described how their experience of building friendships, bonding with peers, and playing in a team during Go Games fostered their enjoyment. For example, one player stated that the best aspect of the Go Games was: "meeting new people and I get to play with my friends". Parents often observed how participating in Go Games allowed their children to bond with teammates and make new friends, recognising how these connections contributed to a positive experience: "I think being with his friends is probably the main thing. If he thinks his friends are all getting up to go out and kick a ball around, he'd come out rain or snow and be happy".

Competition was also a significant source of enjoyment for older players: "just the competitiveness of both teams. It made it very enjoyable". Parents also recognised the inherent competitiveness within their children and its impact on their enjoyment: "For her, she's very competitive so she enjoys that part of it. It's the competition". Supportive of this, the development of competence seemed a prominent source of enjoyment for a substantial number of players engaged in Go Games. Players consistently reflected on a sense of achievement when performing skills, something that was central to their continuing enjoyment: "I'm learning new things, like I'm learning how to do more airstrikes. I wasn't very good at them at the start". This perspective was reiterated by parents, as they observed their children progressing in their skills, leading to a sense of accomplishment. As one parent stated: "he sees that he's improving, that he's getting better and so he likes it, and he always wants to come back".

### 3.3.2. Competition and Competitiveness

Within the context of Go Games, there is no league or championship structure, instead games are organised within a blitz structure where scores are not kept, all players compete, and equal participation is mandated. However, it was evident that even within a non-competitive match structure, there was considerable variation in players' perceptions of the competition outcome. Some players regularly tracked the scores of their games: "the other team had a goal and three points. And our team had about seven or eight goals and three or four points". Moreover, some players placed significant emphasis on winning, primarily highlighting its positive impact on their development, confidence, and motivation: "it makes you play better and grow in confidence". Parents also described how winning, as an important aspect of sport, was important to their children, something that seemed more significant in older age groups: "I think winning is becoming increasingly important for her as the years go past". Yet, this was far from a universal perception with a substantial number less concerned with winning: "I wouldn't really care about scores. I don't care if we win or lose". Interestingly, while many players might not always know the exact score, it was evident that most players had an awareness of whether they won or lost in their

games. This presents a complex picture. Whilst some children enjoyed participation for its own sake, a significant number of participants were highly competitive and driven by the winning and losing of games.

It was evident across both players and parents that the policy of not recording scores did not negate the presence of competitiveness within the Go Games. This is particularly evident in the responses from players, who, when asked to gauge the level of competitiveness in their Go Games on a scale of one (low) to five (high), consistently rated it at four or five. Players expressed that competitiveness within the Go Games was primarily driven by evenly matched teams: "the other team were really good and then there's a lot of people on our team that are good too. So, the teams were evenly matched". This sentiment was echoed by parents who reflected on the need for evenly matched teams to ensure competitiveness and discussed the positive impact of streaming: "they're starting this year to stream them. So, as a result of that, they are competitive because then they're playing against another team that might be at the same level and they're equal". Additionally, parents discussed the competitiveness of their children: "he enjoys the competitive element. He does have a bit of the competitive nature in him". Players and parents also consistently highlighted the importance of competitiveness within Go Games. Notably, competitiveness was seen as a facilitator for improvement and skill development, with one parent stating: "I think you need a certain amount of competitiveness in the game so that they can practise and develop their skills".

## 4. Discussion

Modified sports programs are proposed to engage children in sport and are designed to develop fundamental motor skills and sport-specific skills for future participation [19]. While research provides support for the effectiveness of modified sports programs in terms of skill acquisition [23], there is a paucity of literature examining key stakeholders' perceptions of such programs. Therefore, this study explored the experiences and perceptions of the players, parents, and coaches of Go Games. Specifically, the intended–enacted–experienced curriculum model [35] was utilised as a framework to gain an insight into the potential benefits and challenges associated with the modified sports program.

The results of this study indicate a sense of coherence between the intended curriculum of Go Games, the perceptions of coaches and parents, and the participants' experience. Specifically, the intended principles of equal participation, enjoyment, skill development, and minimal focus on winning are consistently supported across these groups. Coaching is recognised as inherently complex, whereby coaches must navigate a range of pedagogic, social, and cultural problems [50–52]. Although the Go Games principles allowed for coaching practice to be guided towards participant needs, the principles themselves did not help coaches navigate the highly challenging dilemmas of practice (cf. [53]). Whilst there was support for the intended principles proposed by the NGBs, the multi-dimensional curriculum model used in this research allowed for a deeper consideration of these dilemmas. In the Go Games context, coaches had to manage often competing agendas. These included coaching players of varying abilities and diverse motivations, whilst ensuring equal participation and aiming for competitive balance for all. In this context, the biopsychosocial complexity involved in meeting all participants' needs is significantly more complex than is faced at other levels of performance [54].

The Go Games emphasis on fostering enjoyment aligns with extensive research which highlights enjoyment as the largest predictor of commitment and long-term participation in sports, while a lack of enjoyment is the most frequently cited predictor of dropout [9]. Whilst playing pivotal roles in participant engagement [17], there is a significant lack of consistency in defining what constitutes enjoyment [55]. Visek et al. [56] proposed four fundamental constructs of enjoyment in children's sport: contextual (involving practice and games), internal (linked to learning, improvement, effort, and mental rewards), social (centred on team friendships, team rituals, and sportsmanship), and external (involving positive coaching, support during games, and rewards). As a further complexity for the

coach and parent aiming to cater to children's needs equally, the participants in this study attributed enjoyment to the full breadth of constructs suggested by Visek et al. [56]. As an example of the associated coaching dilemmas, Visek et al. [56] suggested that the most influential determinant of enjoyment was the contextual construct which encompassed adequate playing time and competition against evenly matched teams. In the context of this study, coaches were fully aware of the significance of equal playing time; however, the implementation of this principle often led to less challenge for 'more able' players and a detrimental effect on competitiveness if teams were uneven. As such, in this instance, the constructs influencing enjoyment (and long-term participation) were often in conflict with one another.

For the lofty aims of Go Games, and youth sport more broadly, to be achieved there is a need to recognise the coaching expertise required to meet these aims and the depth of pedagogic, curriculum, and 'ology' knowledge required for coaches to be effective [57]. Contrary to the predominant discourse critiquing coach education in the previous literature, we suggest it is important to acknowledge that more, or different, content in coaches' education is unlikely to prepare coaches for these issues [58]. Instead, we suggest the need for greater understanding of the expertise in early-stage participant coaching and opportunities for coaches to work with the complexity of problems found in the present study and others in youth sport [14].

While acknowledging the progress made by NGBs across Gaelic games in implementing the Go Games at the policy level and the impact of these changes, beyond deeper issues related to coaching practice, it would suggest that enhanced communication may help with specific procedural issues and promote shared understanding amongst stakeholders. Specifically, there seemed to be a gap in understanding regarding the purpose of game modifications amongst parents. In particular, coaches were impacted by negative parental involvement, such as shouting instructions or criticisms from the side-lines. Research has consistently demonstrated the significant impact that parents have on sport participation, enjoyment, and development [59]. Yet, this involvement is complex [60], as parents will have a diverse range of experiences within and beyond sport and differing perceptions of behaviours that are appropriate in the youth sport context [59]. Previous research indicates that parental beliefs about youth sport and achievement change significantly as they become more integrated in the sports environment and culture [61,62]. Furthermore, a growing body of research has highlighted the potential impact of interventions, education programs, and workshops on parental involvement in youth sport [63–65]. Therefore, to improve parental understanding and involvement, NGBs and clubs should consider developing parent support programs or resources, with a focus on explaining what to expect and why the intended curriculum is shaped as it is.

The findings of the present study must be considered within the limitations of its design. Firstly, the participants in this study were self-selected volunteers, and their willingness to participate may introduce a potential self-selection bias. Participants who voluntarily agreed to participate in this research may possess unique perspectives or experiences compared to those who declined. There is a risk, of course, that this sampling bias could affect the overall representativeness of our findings, as those who chose to participate may hold stronger views or experiences related to Go Games. Secondly, participants' responses may be influenced by social desirability bias, where they provide answers they believe to be acceptable, rather than expressing genuine thoughts and experiences, concealing critical viewpoints. Lastly, this study employed short semi-structured interviews as the primary method for data collection. While these interviews allowed for a large-scale, focused exploration of key themes and perceptions, the brevity of the interviews may have limited the depth of responses and prevented participants fully elaborating on their experiences. Yet, we believe it to be a strength of this study that the short interviews were conducted 'in situ' as a means of maximising ecological validity.

## 5. Conclusions

In conclusion, the present study examined the experiences and perceptions of players, parents, and coaches within the context of the Go Games modified sports program. The findings highlight the coherence of coaches and parents with the intended core principles of Go Games, emphasising equal participation, enjoyment, skill development, and de-emphasising scores and winning. However, challenges in effective communication with parents regarding the program's purpose and objectives emerged. While parental involvement is crucial to a child's sports experience, it is essential to address the gap in understanding among parents to ensure they positively contribute to the sports environment. Players' enjoyment is influenced by various factors, including social interaction, skill development, competence, and competition. Notably, coaches play a significant role in balancing these elements to create enjoyable experiences. Streaming was identified as an effective approach to ensure competitiveness, particularly for players with varying skill levels. Nevertheless, coaches require better education, support, and tools to make effective development-focused decisions within the modified sports program.

**Author Contributions:** Conceptualization, K.G., J.T., Á.M., S.B. and P.H.; methodology, K.G., J.T., Á.M., S.B. and P.H.; data collection. K.G.; data analysis, K.G., J.T., Á.M. and S.B.; writing—original draft preparation, K.G.; writing—review and editing, K.G., J.T., Á.M., S.B. and P.H.; project administration, J.T., Á.M., S.B. and P.H.; funding acquisition, J.T., Á.M., S.B. and P.H. All authors have read and agreed to the published version of the manuscript.

**Funding:** This research was conducted with the financial support of Science Foundation Ireland [12/RC/2289_P2] at Insight the SFI Research Centre for Data Analytics at Dublin City University, and the Gaelic Athletic Association. For the purpose of Open Access, the author has applied a CC BY public copyright licence to any Author Accepted Manuscript version arising from this submission.

**Institutional Review Board Statement:** This study was completed in accordance with the Declaration of Helsinki and approved by Dublin City University (#2023/135).

**Informed Consent Statement:** Informed consent was ascertained from interested coaches and parents, while informed assent was also obtained from parents on behalf of their children.

**Data Availability Statement:** The data presented in this study are available upon request from the corresponding author. The data are not publicly available due to confidentiality.

**Conflicts of Interest:** Author Jamie Taylor was employed by the company Grey Matters Performance Ltd. The remaining authors declare that the research was conducted in the absence of any commercial or financial relationships that could be construed as a potential conflict of interest.

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
