# Peer review of "Participant Experience of a Modified Sports Program—A Curriculum Investigation in Gaelic Games"

_2673-995X, doi:10.3390/youth4010002_

Round 1

Reviewer 1 Report

Comments and Suggestions for Authors

Thank you for the opportunity to review the study.

The presented study deals with the perceptions of players, parents and coaches of the "Go Games" modified sport programs. This is a modified version of the Gaelic Games that are widespread in Ireland.

Even though this sport is little known across countries, the aim of the study is certainly worthwhile. However, the study presented is not yet ready for publication in this form.

The introduction is very verbose and difficult for the reader to understand. It would be important to start by introducing the Gaelic Games, as it can be assumed that most readers are not familiar with this sport. The resulting problems that led to the Go Games should be explained in a nutshell.

In a second step, the rule variants of the Go Games and the background to them should be explained. The intended-enacted-experienced curriculum model is not familiar to the average reader and should be further explained and specified.

The methods section is difficult to understand. What questions were the interviewees asked? 

The results should not only be presented in narrative form, but also in a structured form (grouping of answers? table form?). At present, the results of the study are not comprehensible.

In general, the study would benefit if the phrasing were more concise and targeted.

Reviewer 2 Report

Comments and Suggestions for Authors

Thank you for the chance to review your paper on modified gaelic games. This aligns significantly with many of my interests, especially since I have conducted much practical and academic work around another modified sport approach known as football3 that aims to encourage inclusion and fair play by re-structuring the game of football. 

In any case, I am generally favorable to this manuscript and have only minor comments to hopefully improve the paper. I provide section-by-section feedback below.

Introduction

Line 49/50: Don't use 'facilitated' twice.

Lines 85: How do Torres and Hager 'not adequately address the diverse range of interests' - it would be good if you expanded on this 

It would be helpful if you expanded slightly on how Go Games modify the rules/equipment/format of Gaelic games. 

Materials and Methods

This is all generally well-described. Two questions I would like to see answered though.

First, how long - on average - were the interviews? 

Second, can you give examples of the kinds of codes you used in the deductive and inductive phases (i.e. what kind of things were you coding for)? 

Results

The presentation of the results strikes me as a bit uneven. 3.1 is quite short compared to 3.2 and 3.3, suggesting perhaps a lack of depth in the 3.1. It is probably worth considering if that section can be expanded or insights from it merged with parts of 3.2/3.3. Indeed, maybe some themes can be combined (e.g. 3.2.3 and 3.3.2) as they are quite similar, even though the 'stakeholder' involved is different. 

Concerning the issue of 'keeping score' in 3.2.3/3.3.2, does your data reveal any times where the scorekeeping became a point of contention despite the curriculum's intention to minimise the scores. You illustrate well how different players/coaches navigate the score and competition issues, but I am wondering if it ever created visible tension or conflict. 

Comments on the Quality of English Language

A minor round of proofing is recommended. 

Reviewer 3 Report

Comments and Suggestions for Authors

I believe that the work is generally correctly done, respecting the general principles of developing a quality research paper.

The introduction - correct, amply presented with the correct delimitation of the terms indicated in the title of the article.

Materials and methods - I recommend the authors -  Participants, Players, according to the average age, to calculate and add the SD value.

At least one illustrative figure should be added to the Results to synthetically express the results obtained following the application of semi-structured interviews.

Discussions, Conclusions, Bibliography - very correctly elaborated!

Round 2

Reviewer 1 Report

Comments and Suggestions for Authors

Dear authors,

Thank you for considering my comments and including them in your manuscript. It is now much easier for me to understand.